# RAIN study: a protocol for a randomised controlled trial evaluating efficacy, safety and cost-effectiveness of intravenous-to-oral antibiotic switch therapy in neonates with a probable bacterial infection

Fleur M Keij,[1,2] René F Kornelisse,[1] Nico G Hartwig,[2] Katya Mauff,[3]
Marten J Poley,[4,5] Karel Allegaert,[1,6] Irwin K M Reiss,[1]
Gerdien A Tramper-Stranders[1,2]

For numbered affiliations see end of article.

**Correspondence to**
Fleur M Keij;
f.keij@erasmusmc.nl

## ABSTRACT

**Introduction** High morbidity and mortality rates of proven bacterial infection are the main reason for substantial use of intravenous antibiotics in neonates during the first week of life. In older children, intravenous-to-oral switch after 48 hours of intravenous therapy has been shown to have many advantages and is nowadays commonly practised. We, therefore, aim to evaluate the effectiveness, safety and cost-effectiveness of an early intravenous-to-oral switch in neonates with a probable bacterial infection.

**Methods and analysis** We present a protocol for a multicentre randomised controlled trial assessing the non-inferiority of an early intravenous-to-oral antibiotic switch compared with a full course of intravenous antibiotics in neonates (0–28 days of age) with a probable bacterial infection. Five hundred and fifty patients will be recruited in 17 hospitals in the Netherlands. After 48 hours of intravenous treatment, they will be assigned to either continue with intravenous therapy for another 5 days (control) or switch to amoxicillin/clavulanic acid suspension (intervention). Both groups will be treated for a total of 7 days. The primary outcome will be bacterial (re)infection within 28 days after treatment completion. Secondary outcomes are the pharmacokinetic profile of oral amoxicillin/clavulanic acid, the impact on quality of life, cost-effectiveness, impact on microbiome development and additional yield of molecular techniques in diagnosis of probable bacterial infection.

**Ethics and dissemination** This study has been approved by the Medical Ethics Committee of the Erasmus Medical Centre. Results will be presented in peer-reviewed journals and at international conferences.

**Trial registration number** NCT03247920

## INTRODUCTION

Antibiotics are among the most prescribed drugs in the paediatric population. With increasing concerns regarding microbial resistance leading to potential multidrug

### Strengths and limitations of this study

► This study addresses an important clinical question regarding intravenous-to-oral antibiotic switch in neonatal care.
► The results of this study can be included in antibiotic stewardship programmes.
► A non-inferiority study design is used.
► The primary outcome assessment is challenging since it is impossible to state whether a new infection episode is a true reinfection since cultures of the first infection were sterile.
► Only probable bacterial infections are included in this study, thus, results cannot directly be applied to culture-proven infection.

resistant infections, antibiotic stewardship programmes have been developed. The main goal of these programmes is optimisation of the use of antibiotics. Moreover, unnecessary use should be reduced since antibiotics do interfere with the development of the gut microbiome in early stages of life, influencing the risk of developing several diseases such as asthma, diabetes and obesity later in life.[1–3] Initially started in adult population, these programmes have, since then, been implemented in the paediatric field, starting with the formation of the Pediatric Committee on Antimicrobial Stewardship (AS) by the Paediatric Infectious Diseases Society in 2010. One of the principles of AS is the appropriate and timely de-escalation of antibiotic use including a switch from parenteral to oral therapy in clinically well children. An early intravenous-to-oral switch leads to many advantages such as a reduction in hospital

stay and associated health costs and is currently described in several guidelines.[4]

However, oral switch therapy is not yet widely practised in neonates. When a bacterial infection is suspected, either based on maternal risk factors or subtle clinical symptoms, antibiotic therapy is immediately started. In the best-case scenario, symptoms disappear, inflammatory parameters are reassuring and cultures remain negative after which therapy can be stopped after 36–48 hours. In some of the neonates (~1% of living births), the presence or absence of a bacterial infection cannot be established.[5] They may have elevated inflammatory parameters and show clinical signs, although (blood) cultures remain sterile; so-called 'probable bacterial infection'. In such cases, antibiotic therapy is usually continued intravenously, up to 7 days.[6]

In low-income and middle-income countries, the above practice is not common due to limited resources. The need for good home-based antibiotic treatment to reduce mortality following neonatal infections has led to several large randomised trials on the use of oral antibiotics. These studies showed that a simplified regimen with a switch to oral amoxicillin after 48 hours is as effective and safe as parenteral therapy.[7–9] Results of those studies have been incorporated in WHO guidelines for the management of neonatal sepsis.[10] Unfortunately, the outcomes cannot be applied to a higher income setting since the studies were based on clinical symptoms solely and no diagnostic tests were performed. This approach has not yet led to further research in high-income settings.

Several small studies have been published on the use of different oral antibiotics, evaluating both pharmacokinetics (PKs) and clinical efficacy. Regarding PKs, studies have commonly found that adequate antibiotic serum levels can be reached following oral administration, although peak levels are lower and appear later in time compared with parenteral administration. Furthermore, a great interindividual variation was seen.[11 12] Small efficacy studies evaluating an early-oral switch in neonates with a possible or proven infection are promising. In a cohort study, 222 infants with possible and proven group B streptococcal (GBS) infection switched to oral amoxicillin in a high dosage after 48 hours of intravenous treatment. During the 3-month follow-up period, no reinfections occurred.[13] Similar results were found in a case–control study in which 36 infants with a possible or proven bacterial infection switched to oral cefpodoxime on day 3. No increase in reinfection rate was observed and the benefits of oral therapy included reduction of hospital stay and an increase in the number of exclusively breastfed newborns.[13 14] Unfortunately, study sizes were small and studies were not randomised. Furthermore, no full cost-effectiveness analyses of paediatric intravenous-to-oral switch therapy have been reported to date. This represents an important knowledge gap, considering that, in the current era of austerity and rising healthcare costs, it

is increasingly required to demonstrate whether or not new treatment strategies offer good value for money.

Together with uncertainties about bioavailability after oral administration, the lack of firm evidence delays further implementation in a high-resource setting. When intravenous-to-oral antibiotic switch therapy in neonatal infections is proven safe and effective, it can lead to a great improvement in quality of life (QOL) for both child and parents. Furthermore, it can reduce the length-of-hospital stay, which will have positive effects on rising healthcare costs.

## Objectives

The primary objective is to demonstrate non-inferiority of intravenous-to-oral antibiotic switch therapy in clinically well neonates with a probable bacterial infection compared with a complete course of intravenous therapy. The secondary objectives are: (1) to describe PKs of oral amoxicillin/clavulanic acid in neonates; (2) to quantify the cost-effectiveness of intravenous-to-oral switch therapy in neonates; (3) to evaluate the QOL in relation to switch therapy in neonates; (4) to study modification of the gut microbiome in relation to type and administration of antibiotics and (5) to explore the additional value of molecular diagnostics on bacterial infection using left-over blood culture material that remains after culture time has passed.

## METHODS AND ANALYSIS
### Study design

The RAIN study (Reduction of intravenous Antibiotics In Neonates) is a multicentre, randomised controlled, open label, non-inferiority trial. Patients will be recruited in 17 large teaching hospitals in the Netherlands. For details on participating hospitals, we refer to the online supplementary material.

## Participants

Neonates are eligible for participation if they have a postmenstrual age of 35 weeks or more, are 0–28 days old, have a body weight of at least 2 kg and have a 'probable bacterial infection' based on clinical symptoms or maternal risk factors and elevated inflammatory parameters for which the paediatrician has decided to complete a full course of intravenous antibiotics. A C reactive protein (CRP) level of ≥10 mg/L is considered as elevated. For procalcitonin (PCT), we refer to the age-adapted ranges.[15 16] Clinical symptoms can be found in the online supplementary material. All neonates with a proven bacterial infection, for example, positive blood or cerebrospinal fluid culture or a severe clinical sepsis requiring intensive care treatment are excluded from participation. Inclusion and exclusion criteria are described in table 1. Participants will be screened by the local paediatrician or investigator and randomised after 48 hours of intravenous treatment once informed consent has been obtained.

**Table 1** Inclusion and exclusion criteria Reduction of intravenous Antibiotics In Neonates study

| Inclusion criteria | Exclusion criteria |
|---|---|
| ► Neonates, postmenstrual age ≥35+0 weeks, postnatal age of 0–28 days, body weight ≥2 kg. <br> ► Probable bacterial infection defined as clinical symptoms and/or maternal risk factors and elevated inflammatory markers for which empiric broad-spectrum antibiotic treatment was initiated and needs to be continued for >48 hours. <br> ► Clinically well. <br> ► Tolerate oral feeding without overt vomiting. | ► Proven bloodstream infection. <br> ► Absence of blood culture. <br> ► Severe localised infection (meningitis, osteomyelitis, necrotising enterocolitis). <br> ► Severe clinical sepsis (compromised circulation, need for mechanical ventilation). <br> ► Continuous need for a central venous line. <br> ► Severe hyperbilirubinaemia exceeding the exchange level. <br> ► Parents' inability to administer medication. |

## Intervention

The intervention group will switch to oral amoxicillin/clavulanic acid after 48 to 72 hours of intravenous antibiotics. The number of intravenous antibiotic doses prior to the switch will be recorded for every participant. The control group will receive the full course of intravenous antibiotics, which consists of penicillin and gentamicin, in case of early-onset sepsis, in most participating centres. Both groups will be treated for a total of 7 days. The trial flow diagram is outlined in figure 1.

### Dosage, method of administration

A daily oral dose of 75/18.75 mg/kg of amoxicillin/clavulanic acid, thus 25/6.25 mg/kg every 8 hours, will be administered. The mode of administration is orally via suspension, or in case of feeding problems (with the exception of vomiting) per nasopharyngeal tube. If the neonate vomits significantly within 30 min after administration, the administration will be repeated.

## Randomisation and blinding

Participants will be randomised after 48 hours of intravenous treatment by a web-based programme according to a stratified block randomisation (stratified per site in alternated blocks of 2 and 4). Randomisation ratio will be 1:1. Randomisation can under no circumstances be influenced by the study team. If parents want their child to be in a specific group, they will be excluded from participation. Patients and healthcare professionals will not be blinded, because of the nature of treatments (intravenous vs oral therapy); the primary investigators performing the statistical analysis will be blinded for the allocated treatment.

## Follow-up

Neonates will only be discharged from the hospital if they have shown to tolerate the oral antibiotic treatment for at least two doses and are clinically well. Parents will be interviewed by telephone on the first day after discharge and on day 14 after initiation of antibiotic therapy by the local study investigator or research nurse. The local investigator will assess antibiotic intake, signs of infection, side effects and events. In addition, parents will receive a survey on day 7 and day 21 with questions regarding possible signs of infection, side effects, QOL, loss of productivity, and cost-effectiveness

and demographic data (maternal, paternal age, highest level of education, socioeconomic status and siblings). All surveys are available as online supplementary material. The primary outcome of the study is assessed 28 days after treatment completion, when the patient will be seen by the local investigator or research nurse for a final visit at the outpatient department. Parents of participants in the oral treatment group will be asked to return the bottle with remaining antibiotic suspension in order to evaluate medication adherence through measurement of the remaining amount of medication present in the bottle. Follow-up of neonates in the control group will be identical.

## Power calculation

Relapse of proven GBS infection is estimated to occur in 0%–8% of patients, but may be caused by a different strain.[17] Two epidemiological studies estimated the recurrence rate to be 1% in proven bacterial neonatal infections.[18 19] The current study focuses on probable infections (in which the recurrence rate is likely even lower, although evidence for this is lacking). The incidence of community-acquired late-onset sepsis in otherwise healthy neonates is low (~0.28 per 1000 live births).[20] Therefore, we assume a 1% infection rate (including new non-related infections) in both groups. Based on a non-inferiority margin of 3% (minimum success rate 96%), 2×231 patients are needed for a power of 90% using a one-sided alpha of 0.025. Because of the first 30 patients undergoing PK measurements and uncertainty about the PK values, and potential drop-out or withdrawal (estimated approximately 5%–10%), we will include 2×275 neonates (total=550). A post hoc report on the design effect will be included.

## Data collection and management

Data will be collected by the local investigator at each site and stored anonymously in a Good Clinical Practice proof digital database (Castor EDC, Cewit B.V). Two independent monitors will perform source data verification and assess performance of study procedures at each site following a predefined schedule.

## OUTCOME MEASURES AND STATISTICAL ANALYSIS
### Primary outcome

The primary outcome is bacterial (re) infection within 28 days after finishing antimicrobial therapy, defined

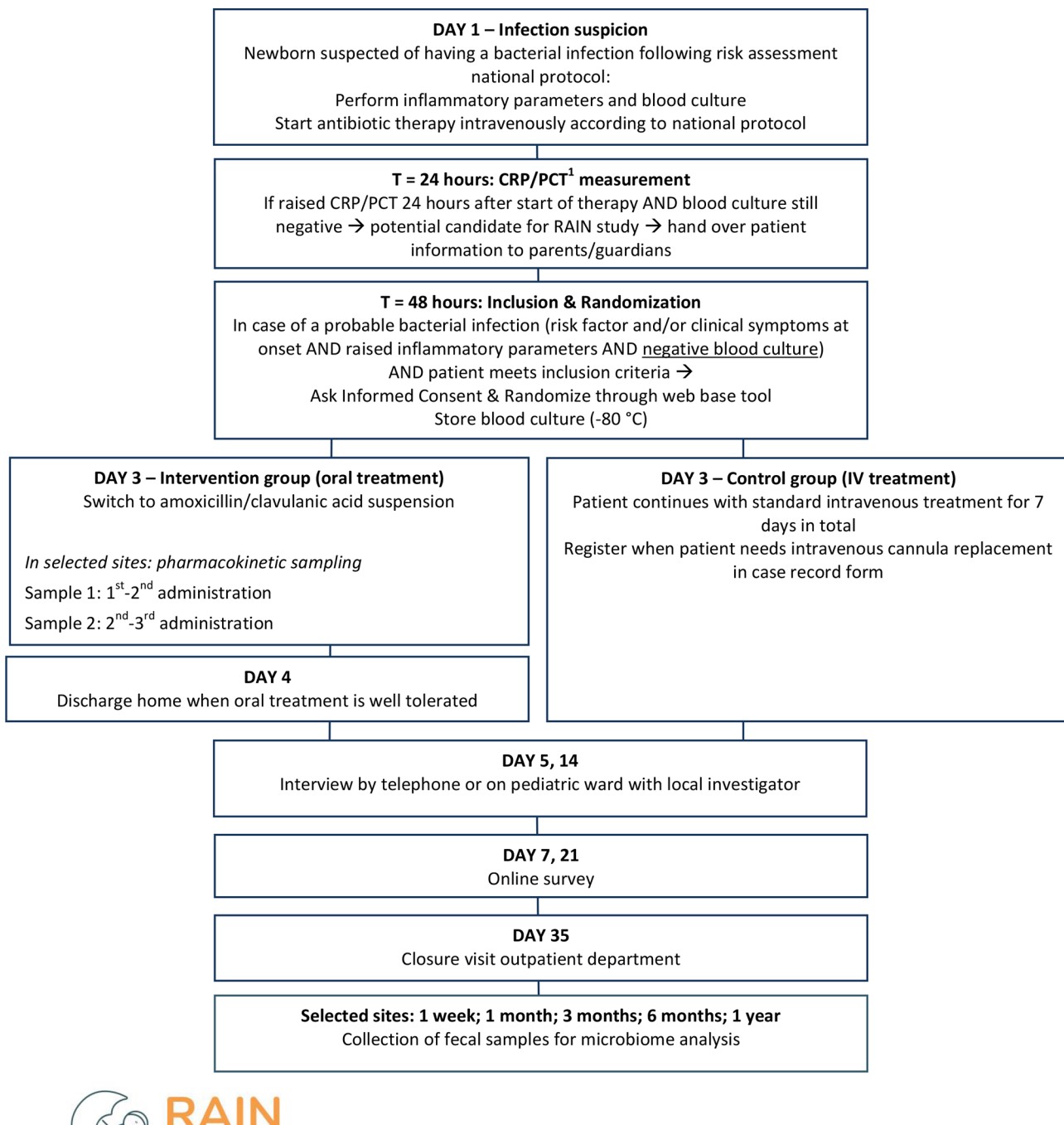

**Figure 1** Flow chart of study procedures. [1]CRP, C reactive protein; PCT, procalcitonin; RAIN, Reduction of intravenous Antibiotics In Neonates.

as clinical symptoms of infection and fever (>38.0°C) or hypothermia (<36.0°C) and elevated inflammatory parameters with a need for prolonged antimicrobial therapy for more than 48–72 hours. Proven or suspected viral infection, for which no antibiotics are initiated or antibiotics are stopped within 48–72 hours (after return of culture and/or viral results) are not considered as a bacterial (re) infection. A chart will aid in decision making with respect to differentiation between bacterial infection and other causes. Both per-protocol analysis and intention-to-treat analysis will be performed. A

p<0.025 will be considered as significant. Regarding the primary outcome, 28-day cumulative bacterial infection rate; proportions will be defined and used to compare both treatment strategies. Subgroup analysis will be performed on late preterm neonates (gestational age 35–37 weeks). Further, mixed-effect models will be used to take into account the multiple centres.

### Interim analysis

A formal interim analysis will be performed after occurrence of the first four primary outcomes. The data safety

monitoring board (DSMB) will assess the results and it will be determined whether the occurrence of the primary outcome was related to the index suspected bacterial infection or more likely to be an unrelated infection. The outcome of that interim analysis will determine after how many primary outcomes the next interim analysis will be performed. When more than 75% of related cases occur in the intervention arm in two consecutive interim analyses, the study will be stopped prematurely.

Secondary outcomes and their statistical analyses are described below.

## Secondary outcomes
### Pharmacokinetics
Blood samples of the first 30 patients treated with oral antibiotics (intervention arm) will be analysed in order to confirm that plasma concentrations are within target range and to endorse safety and dosing of the investigated product. It is thought that time above the minimal inhibitory concentration (MIC) should be at least 40%–50% in order for treatment with penicillins to be adequate.[21] Therefore, T>MIC should be at least 50% in 90% of the cases to be considered adequate, otherwise dosing needs to be adjusted. A target MIC of 8 mg/L is chosen for amoxicillin since this is the clinical breakpoint for *Escherichia coli* susceptibility defined by the European Committee on Antimicrobial Susceptibility (Eucast).[22] In a later stage, sampling will be continued in order to develop a pharmacokinetic/pharmacodynamic (PK/PD) model and to perform a population PK analysis. For the first 30 patients, two blood samples (0.5 mL whole blood) will be collected; the first sample 4–6 hours after the first or second dose, the second sample 6–8 hours after the second or third dose of oral suspension. In a subset of patients, we will collect two samples following the same administration in order to calculate an area under the curve. After the first analysis on samples from 30 patients, sampling will be continued through the collection of a single blood sample per patient. Samples will be handled within 1 hour after collection by the local laboratory. The sample will be centrifuged at 3000 rpm for 10 min and the recovered plasma will be stored at −80°C. Once enough samples have been collected for analysis, they will be shipped to the Erasmus MC for analysis. Serum concentrations of both amoxicillin and clavulanic acid will be measured. For each measurement, 50 µL of plasma is needed. Analysis will be performed batchwise combining both liquid chromatography and mass spectrometry. Analysis will be performed by descriptive statistics and by using non-linear mixed-effects modelling including variables such as timing of blood drawing, timing of last meal, patient weight, centre.

### Molecular techniques for bacterial detection
Left over material of blood culture before initiation of antibiotic treatment will be stored at −80°C. Molecular analysis will be performed using next generation sequencing (NGS) methods, evaluating the additional yield of those techniques on bacterial detection compared with blood culture. Analysis will be performed batchwise, meaning that the additional value will be determined afterwards and that results will not be immediately available for the treating clinician nor for the DSMB in case of a (re)infection. However, results will be available at the end of the study and will be analysed in regard to primary outcomes that have occurred during the course of the study. The analysis of the NGS will initially be exploratory. If substantial differences are noted in the yield of recovered organisms from NGS compared with the blood culture, further analysis may be undertaken, exploring the relationship with clinical data such as inflammatory parameters and clinical illness. In that instance, mixed models will be used, and adjustments will be made for centre.

### Gut microbiome
Microbiome alterations following antibiotic therapy, both in the intervention and control group, will be evaluated in 80 participants from four centres. Faecal samples will be collected at five different moments in the first year of life (around day 7, 30, 90, 180 and day 365) from neonates born vaginally. Samples will be collected from the diaper and immediately stored at −20°C in a special box placed in the home freezer. Parents will receive a notification at each sampling moment including a questionnaire regarding episodes of antibiotic treatment, type of feeding, daycare attendance and the presence of siblings at home. After 365 days, all samples will shipped to the central research locations and stored at −80°C until analysis in a specialised facility using high throughput sequencing targeted at the bacterial 16S rRNA locus. Alpha diversity of all samples will be calculated by the Shannon index. We will identify time points in which the alpha diversity and relative abundances in operational taxonomical units differ between the infants with intravenous treatment and infants with intravenous-to-oral switch with a smoothing spline analysis of variance. Moreover, differences in the relative abundances of the genera will be tested with a linear mixed-effect model with correction for the variables of other antibiotic use, feeding type, day care attendance/sibling and centre.

### Cost-effectiveness analysis
Following a superiority design, the cost-effectiveness analysis (CEA) will make a comparison between the intervention and the control group by identifying, measuring and valuing the costs and patient outcomes of both treatment strategies.

Following established methods for economic evaluations and costing studies in healthcare, both medical costs and non-medical costs (ie, costs of productivity losses) will be analysed for both study groups.[23 24] Total medical costs over the 28-day follow-up period will be calculated, including costs of intravenous antibiotics, costs of oral antibiotics, laboratory costs and costs of hospitalisation days. In regard to readmission costs, only costs of a readmission for a bacterial (re) infection, as defined in the

outcome measures section, will be added to the analysis. Resource consumption will be derived from electronic hospital databases. Unit prices will be calculated using real economic cost prices or using standard cost prices for health economic evaluations. Unit prices will be multiplied by the quantities for each resource used, and then summed over the separate types of resource to give a total cost per patient. Then, productivity costs in the patients' parents/caregivers, consisting of productivity losses regarding both paid and unpaid work, will be analysed (ignoring prearranged, regular maternity and paternity leave). These costs will be measured using the institute of Medical Technology Assessment Productivity Cost Questionnaire,[25] completed by the parents during the scheduled follow-up as described above. The productivity costs will be valued following the friction cost method. Briefly, the valuation of paid work will be based on average hourly labour costs per paid employee, whereas that of unpaid work will be based on the replacement costs for household work. Finally, all costs will be summated for each individual patient.

Regarding the patient outcomes, the CEA will look at the number of bacterial (re) infections within 28 days after finishing of antimicrobial therapy. Building on the data on costs and outcomes described above, a cost-effectiveness ratio will be calculated (the final endpoint of the CEA). This ratio will be expressed as incremental costs per bacterial (re)infection avoided. Otherwise, the CEA will focus on dominance of one treatment over the other with respect to lower cost and greater effect. Analysis of uncertainty in the estimation of the incremental cost-effectiveness ratio is illustrated through a cost-effectiveness plane (via bootstrapping). Where relevant, sensitivity analysis will be performed to assess the robustness of the analysis to certain assumptions.

### Quality of life
To get an impression of the impact of an early intravenous-to-oral antibiotic switch on the QoL of both parents and their child, parents will be asked to fill in a questionnaire on day 7 and day 21. Since there is no validated QoL instrument for neonates <1 month, we developed a questionnaire ourselves that collects information on total admission days and readmission rate, symptoms, side effects, amount of intravenous cannulas inserted during admission, sleep quality of the patient, number of medical visits and medication use in the first month, feeding and breastfeeding success rate and parents' satisfaction. Outcome reporting will be mainly descriptive, if a trend is observed for some of the questions, mixed-effect models can be used to further explore the effect of treatment and centre.

### Safety consideration
Serious life events (SAEs) will be reported to the principal investigator within 48 hours. The principal investigator will report the adverse events to the Medical Ethics Committee of the Erasmus Medical Centre and the Central Committee on Research Involving Human Subjects. All SAEs will be followed until they have abated, or until a stable situation has been reached. An independent DSMB will be installed consisting of an epidemiologist/statistician, consultant neonatologist and paediatric infectious diseases specialist. The DSMB will evaluate the progress of the trial and will examine safety parameters at yearly intervals and through the performance of interim analyses when necessary. The adverse events will be listed and discussed with the DSMB.

### Patient and public involvement
The Dutch parent organisation 'Vereniging van Ouders van Couveusekinderen (VOC)' was involved in the conceptualisation of the trial and has given their approval for the execution of this study. Patients will be asked at inclusion if they want to be informed on the final study results by email. Furthermore, a summary of findings will be published on the website of the VOC and on our own website.

## DISSEMINATION
The results of the primary and secondary aims will be published in peer-reviewed journals and presented at international conferences, followed by a press release.

## DISCUSSION/CONCLUSION
Prolonged hospitalisation for intravenous therapy is still common practice in high-income societies when neonatal bacterial infection cannot be ruled out despite negative cultures. In-hospital treatment has a high impact on both patient and parents since prolonged hospitalisation often leads to infant–parent separation. Initiatives to prevent separation through the reduction of admission days are of increasing interest. The additional value of other biomarkers besides CRP, such as PCT, in order to rule out bacterial infection has been evaluated. A recent study showed that using PCT values as guidance for antibiotic therapy in neonates with a suspected bacterial infection leads to a reduction in days on therapy and hospitalisation. However, when the infection could not be ruled out, as is the case in infants with a probable infection, therapy was still continued intravenously for 7 days.[26] We propose the first randomised controlled trial in a high-resource setting with a non-inferiority study design in order to demonstrate that oral antibiotic switch therapy is as effective as intravenous treatment in neonates with a probable bacterial infection. Furthermore, we will be the first to perform PK/PD analysis on oral amoxicillin/clavulanic acid on a large sample size. Once proven that serum concentrations following oral administration are adequate, and non-inferiority of oral switch has been proven, oral switch therapy could potentially be expanded to clinically well infants with a proven infection.

This study has some limitations, the most important limitation relating to the primary outcome assessment. As

mentioned in the methods section, only neonates with a probable bacterial infection will be included. It is impossible to state whether or not a second infection is a reinfection or that it is caused by a different pathogen. We, therefore, developed a list of recommended diagnostics that should be determined when a participant returns with signs of a possible infection. Those results can be used for correct and uniform assessment of the event and will help differentiating between a true bacterial infection, a probable bacterial infection or a viral infection.

Another limitation of the study relates to the fact that PK blood sampling will be performed in the intervention group. As this procedure carries a small risk of infections, bias may be introduced into the study. On the other hand, patients in the control group are at risk for infection when replacement or new insertion of an intravenous cannula is needed. Both situations can lead to a possible bacterial infection. We think, however, that the infection risk is low when performed correctly including good hygiene measurements. Moreover, neonates in both groups are under antibiotic treatment which also covers *Staphylococcus aureus*.

Finally, bias may occur because of the oral treatment being administered out of the hospital setting. Since this treatment is home based, parents' compliance in administering medication may be suboptimal, and we might underestimate the effectiveness of the intervention. In an attempt to overcome this limitation, infants will only be discharged when parents have shown to adequately administer the antibiotic suspension, parents will be phoned 1–2 days after discharge to evaluate medication administration, and parents will be asked to return the bottles with remaining antibiotic suspension at the final visit (which allows us to estimate the volume of the suspension actually administered).

We assume that once effectiveness and safety have been shown, implementation of intravenous-to-oral switch therapy will reduce the risks of hospital-related complications such as nosocomial infections and lead to an earlier return to a normal family setting with improvement in QOL and better mother–child bonding. Furthermore, we expect that the cost reduction will be substantial.

**Author affiliations**
[1]Pediatrics, Division of Neonatology, Erasmus MC-Sophia Children's Hospital, Rotterdam, The Netherlands
[2]Pediatrics, Franciscus Gasthuis & Vlietland, Rotterdam, The Netherlands
[3]Biostatistics, Erasmus Medical Center, Rotterdam, The Netherlands
[4]Pediatric Surgery, Erasmus MC-Sophia Children's Hospital, Rotterdam, The Netherlands
[5]Medical Technology Assessment (iMTA), Erasmus University Rotterdam, Rotterdam, The Netherlands
[6]Department of Development and Regeneration, KU Leuven, Leuven, Belgium

**Acknowledgements** We thank The Dutch parent organisation 'Vereniging van Ouders van Couveusekinderen (VOC)' for their help in the conceptualisation of the trial.

**Contributors** GAT-S conceived the original idea for this trial. FMK, RFK, KM, MJP, NGH, KA and IKMR were involved in further conception and trial development. FMK wrote the first draft. FMK, GAT-S and RFK rewrote the article. GAT-S, RFK, KM, MJP, NGH, KA and IKMR critically revised the article for important intellectual content. All authors contributed to and approved the final version of the manuscript.

**Funding** This work is supported by The Netherlands Organisation for Health Research and Development (ZonMW) grant number 848015005 and the Innovatiefonds Zorgverzekeraars.

**Competing interests** None declared.

**Patient consent for publication** Not required.

**Ethics approval** This study was approved by the Medical Ethics Committee of the Erasmus Medical Centre (NL51888.078.16, June 2017). The study will be conducted according to the principles of the Declaration of Helsinki (seventh revision, October 2013) and in accordance with the Medical Research Involving Human Subjects Act (WMO) and other guidelines, regulations and Acts such as Good Clinical Practice.

**Provenance and peer review** Not commissioned; externally peer reviewed.

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
