## [Reviewer comments · BMJ Open]

ARTICLE DETAILS

TITLE (PROVISIONAL)	RAIN study: a protocol for a randomized controlled trial evaluating efficacy, safety and cost-effectiveness of intravenous-to-oral antibiotic switch therapy in neonates with a probable bacterial infection
AUTHORS	Keij, Fleur; Kornelisse, René; Hartwig, Nico; Mauff, Katya; Poley, Marten; Allegaert, Karel; Reiss, Irwin; Tramper-Stranders, Gerdien

VERSION 1 – REVIEW

REVIEWER	Dr Gabriella Morley University of Birmingham, United Kingdom
REVIEW RETURNED	11-Nov-2018

GENERAL COMMENTS	Thank you to the authors for this study protocol entitled: Intravenous to oral antibiotic switch therapy for suspected neonatal bacterial infections: a randomised controlled trial (RCT) to study clinical efficacy, safety, and cost effectiveness. This protocol describes the plan for an RCT which aims to address the question of whether an early (at 48 hours) switch from IV to oral antibiotic therapy, in suspected neonatal bacterial infections, has no effect on the reinfection rate of neonates. Further objectives of this study include: an analyses of the cost effectiveness of oral compared with IV therapy; quality of life, pharmacokinetics of oral therapy and microbiome development. Strengths of this study protocol 1. Firstly, the study protocol details an RCT which should help to address the paucity of evidence currently available with regard to antibiotic treatment regimes in neonatal patients.2. The protocol adheres to the CONSORT 2010 recommendations, where possible, at this stage prior to the RCT. Limitations of this study protocol 3. One of the major limitations of the RCT described in this protocol, which the authors have not themselves addressed, is the assessment of whether the primary outcome, bacterial reinfection, has occurred. Reinfection has not been defined within this protocol, however one would assume that the standard definition of reinfection, that being, a second infection that follows recovery from a previous infection by the same causative agent, is used in this study. Given the participants of this study, by definition, have sterile bacterial culture, it is therefore not possible to determine whether a true reinfection with the same causative agent has actually occurred. That being said, given the primary outcome is the same in both groups and participants in each group have
--

sterile cultures on inclusion in the study, interpretations can be drawn in comparing the groups in regard to any subsequent infection after the intervention period. This subsequent infection could be hypothesised as being a reinfection but the authors should clearly state in this study protocol that this can only be a hypothesis and not a concrete conclusion.

4. The limitations of the RCT itself are not fully addressed with only one limitation being described on page 3. The authors should provide a more in depth analysis of the integral limitations with this RCT as this is required to demonstrate how the possible future results would need to be interpreted.

Abstract

5. Please clarify in the abstract on line 37 (page 2) whether it is treatment completion or treatment initiation: "the bacterial reinfection within 28 days after treatment."

6. Please include all secondary outcomes in the abstract: it appears that new diagnostic strategies is not included in the abstract.

Introduction

7. More detail with regard to other evidence for early IV to oral switch in paediatric patients would be advantageous. This has been eluded to in the abstract but not developed in the introduction.

Methods and Analysis

8. Please provide further detail of the locations and hospitals involved the trial.

Primary outcome

9. Please further detail the inclusion criteria with specific emphasis on the raised CRP/PCT and clinical symptoms. For example what defines a raised CRP/PCT in this RCT? What clinical symptoms/signs define probably bacterial infection in this RCT? It may be appropriate to have a checklist of symptoms and to define how many of these symptoms or which symptoms in particular are necessary for the neonate to be included in the study.

10. With regard to this, in figure 1, the flow diagram, please be specific with regard to wording see box T=48 hours: the brackets defining the probable infection should relate to the text (on page 7 line 20) and perhaps therefore should state clinical symptoms AND fever or hypothermia AND raised inflammatory parameters.

Secondary outcomes

11. Throughout the manuscript there are discrepancies with regard to what is included in the secondary outcomes. Please address this and make uniform throughout. For example, on page 7 line 31 it states that duration of hospitalisation will be an included secondary measure, however this has not been stated elsewhere in the manuscript.

12. Randomisation after 48 hours is the critical point in the RCT however, one suspects that in clinical practice there may be variation in how quickly participants are randomised to each group. This will not be an issue for the control group however it may be that certain participants in the intervention arm have more doses of

	IV antibiotics if the 48 hour switch to oral is not performed optimally in all participants. This should be considered by the authors and perhaps it would be useful to collect data with regard to the number of doses of both IV and oral antibiotics for each participant in the intervention arm, to record any deviations from the optimum 48 hour switch. 13. Given the RCT is not blinded to caregivers, parents and clinicians, the authors should address the standard procedure for if a parent/s wants their child to be in a specific group for the trial. 14. The authors should consider the bias introduced by oral treatment being administered out of the hospital setting, to include some consideration of drug dose, duration and frequency compliance. 15. Please could the authors include further details of the outcome survey to include the questions on signs of infection, side effects, quality of life and cost effectiveness. See page 9 line 12. 16. Page 9, line 14 states the primary outcome in a different format to previously described (5 weeks, as opposed to 28 days), please standardise across the manuscript. 17. Please could the authors ensure that all acronyms are adequately explained in full, for example GBS on page 9 and DSMB on page 9. And CEA on page 12. 18. The DSMB are the allocated group who are to determine whether the primary outcome is a reinfection. It is important to state what factors will be considered in order to come to this decision. Page 9 19. The timing of the pharmacokinetic blood sampling, after the first 30 patients, is not defined. One assumes this is because the first 30 patients will establish the optimum time for blood sampling. However, it should be considered that the participant may be discharged home after the second dose of oral antibiotic, if tolerated. Therefore, if the optimum time for blood sampling is after the 3rd dose of oral antibiotic, this potentially prolonged hospital stay needs to be justified or alternative procedure should be put in place for the blood sampling. 20. With regard to the pharmacokinetic blood sampling, the authors should consider how this is a variable introduced into the intervention group but not the control group. Blood sampling is an invasive procedure and carries a small risk of introducing infection. the authors should consider how this may introduce bias and how this could be controlled for or at least acknowledged as a limitation. 21. The authors should state what demographic data will be collected to ensure the analysis and comparisons between groups is matched or controlled for. 22. Storage of left over blood material. Please outline when the NGS will be conducted. In addition, please outline whether the NGS data will be available for DSMB when assessing whether a second infection is a reinfection or not.
--	--

	23. Please provide as supplementary figure the gut microbiome questionnaire. 24. Please detail whether the microbiome data will be collected from the intervention arm only or both. The flow diagram seems to suggest both, however the text on page 11 does not. 25. Please detail whether control microbiome samples will be collected for example from neonates with no antibiotic treatment. 26. With regard to cost effectiveness and productivity cost, please detail how the analysis will take into account prearranged maternity and paternity leave. 27. Please provide further details of the quality of life questionnaire perhaps as a supplementary figure. 28. For the quality of life questionnaire the authors should consider whether questions with relation to parental anxiety may be important to include. For example, with regard to whether a certain type of intervention is more anxiety inducing and/or parental anxiety about future bacterial infection with relation to each intervention. 29. Ethics and dissemination – it may be important to consider whether there should be dissemination to the wider public and certainly public engagement with regard to this RCT. Given the importance of antibiotic stewardship, this could offer an opportunity for public engagement and discussion about antibiotic usage and consumption.
--	--

REVIEWER	Celia Cooper Women's and Children's Hospital, North Adelaide, South Australia, AUSTRALIA
REVIEW RETURNED	20-Nov-2018

GENERAL COMMENTS	A very worthwhile study that has the potential to collect extremely useful data relating not only to clinical management but also to the PK of oral antibiotics in neonates and the impact of oral antibiotics (especially a broad spectrum antibiotic such as amoxicillin/clavulanic) on the developing gastrointestinal microbiome. These are all questions that need answering to advance the practice of neonatal AMS. The role of WGS as complementary to blood culture in cases of neonatal culture negative sepsis is also important to define. I would like to know why amoxicillin/clavulanic was chosen as the oral comparator. Do the authors have data on blood culture-proven neonatal sepsis that suggests that the spectrum is appropriate? The dose of amoxicillin/clavulanic is higher than we use at our institution as we would give 22.5mg/kg bd rather than tds. This seems like a very high dose to me and might lead to a higher rate of adverse events. Regarding the formal interim analysis - how will the DSMB determine whether the primary outcome is related to the index suspected bacterial infection or not? Since the index infection has not been proven (as a condition of inclusion of the patient in the trial) how can a determination be made regarding the relatedness of a second infection? Some more details here would be useful since this is the primary safety assessment for early cessation of the trial.
--

	Is there any other way of collecting blood from neonates for PK studies other than by collecting additional samples e.g. scavenging blood collected for routine clinical purposes? Regarding microbiome alterations, why are samples only being collected from the patients receiving augmentin? Should samples be collected from the standard therapy patients as well?
--	--

REVIEWER	Rahim Moineddin University of Toronto, Canada
REVIEW RETURNED	26-Dec-2018

GENERAL COMMENTS	Comments on 'Intravenous-to-oral antibiotic switch therapy for suspected neonatal bacterial infections: a randomized controlled trial to study clinical efficacy, safety, and cost-effectiveness RAIN study: Reduction of intravenous Antibiotics In Neonates' Authors presented a protocol for a balanced multicenter (17 large teaching hospitals in the Netherlands) randomised controlled trial assessing the non-inferiority of an early intravenous-to oral antibiotic switch compared to a full course of intravenous antibiotics in neonates (0-28 days of age) with a probable bacterial infection. The sample size is calculated for randomized controlled trials not a multicenter randomized controlled trial. The multicenter nature of the study is ignored. Authors must estimate the design effect and multiply the calculated sample size (550) by the design effect. The suggested statistical methods for analyzing the secondary outcomes (secondary outcomes will be analyzed using a t-test or Mann- Whitney U test and in the case of categorical data a Chi-square test will be used) are not appropriate because the multicenter nature of the study is ignored. Authors are familiar with the mixed effects models, however for both primary and secondary analyses mixed effect or Generalized Estimation Equation methods must be used.
--

VERSION 1 – AUTHOR RESPONSE

Reviewer 1

1. Firstly, the study protocol details an RCT which should help to address the paucity of evidence currently available with regard to antibiotic treatment regimes in neonatal patients.
2. The protocol adheres to the CONSORT 2010 recommendations, where possible, at this stage prior to the RCT.

Thank you to dr. Morley for critically reviewing our manuscript. We are pleased to read that she considers this to be a relevant study and we are happy with the comments given to further improve the study protocol.

3. One of the major limitations of the RCT described in this protocol, which the authors have not themselves addressed, is the assessment of whether the primary outcome, bacterial reinfection, has occurred. Reinfection has not been defined within this protocol, however one would assume that the standard definition of reinfection, that being, a second infection that follows recovery from a previous infection by the same causative agent, is used in this study. Given the participants of this study, by definition, have sterile bacterial culture, it is therefore not possible to determine whether a true reinfection with the same causative agent has actually occurred. That being said, given the primary outcome is the same in both groups and participants in each group have sterile cultures on inclusion

in the study, interpretations can be drawn in comparing the groups in regard to any subsequent infection after the intervention period. This subsequent infection could be hypothesised as being a reinfection but the authors should clearly state in this study protocol that this can only be a hypothesis and not a concrete conclusion.

We agree with the reviewer that it is impossible to state whether an infection is a real re-infection or not because of earlier sterile cultures. Therefore, we will clarify this both in the primary outcome section on page 11 and write a statement in the discussion/limitations on page 18. We will also rephrase re-infection to (re)-infection. Our aim is to compare the infection rate within the first 28 days after treatment completion between the groups, which indeed might be composed of true re-infections with a pathogen that has never been known or new, different infections.

4. The limitations of the RCT itself are not fully addressed with only one limitation being described on page 3. The authors should provide a more in depth analysis of the integral limitations with this RCT as this is required to demonstrate how the possible future results would need to be interpreted. We incorporated the limitations mentioned in comment 3 in the discussion of the manuscript and added some other limitations. See page 18 of the manuscript.

Abstract

5. Please clarify in the abstract on line 37 (page 2) whether it is treatment completion or treatment initiation: "the bacterial reinfection within 28 days after treatment."

Treatment completion, corrected in manuscript on page 2

6. Please include all secondary outcomes in the abstract: it appears that new diagnostic strategies is not included in the abstract.

Corrected in manuscript on page 2

Introduction

7. More detail with regard to other evidence for early IV to oral switch in paediatric patients would be advantageous. This has been eluded to in the abstract but not developed in the introduction. Data on oral antibiotics and antibiotic switch therapy in neonates are scarce. In the introduction, we have mentioned the WHO studies, and also few other studies performed in high-income settings. We have expanded on these studies in the introduction, also mentioning their strengths and limitations.

Methods and Analysis

8. Please provide further detail of the locations and hospitals involved the trial.

A list of all participating hospitals has been added to the supplementary material.

Primary outcome

9. Please further detail the inclusion criteria with specific emphasis on the raised CRP/PCT and clinical symptoms. For example what defines a raised CRP/PCT in this RCT? What clinical symptoms/signs define probably bacterial infection in this RCT? It may be appropriate to have a checklist of symptoms and to define how many of these symptoms or which symptoms in particular are necessary for the neonate to be included in the study.

All participating hospitals use the Dutch national guideline on Early Onset neonatal sepsis which has been adapted from the UK Nice guidelines. In this guideline, a CRP of 10 mg/L or more is considered to be elevated. For PCT reference values the age-related nomogram is used. (1)

Apart from elevated inflammatory parameters, maternal and neonatal risk factors and/or clinical signs of infection have to be present in order for the newborn to be included, as mentioned in the inclusion criteria listed in table 1. Risk assessment tables for suspected early onset bacterial infection from the national guideline are included in the supplementary material of the manuscript. These tables are slightly modified from the tables of the NICE guidelines. (2) We do register clinical symptoms at onset

of infection suspicion, maternal and neonatal risk factors and inflammatory parameters in our database. Comparison of baseline data between centers/regions is therefore possible.

10. With regard to this, in figure 1, the flow diagram, please be specific with regard to wording see box T=48 hours: the brackets defining the probable infection should relate to the text (on page 7 line 20) and perhaps therefore should state clinical symptoms AND fever or hypothermia AND raised inflammatory parameters.

As mentioned under 9, Infants do not necessarily have to show clinical signs of infection in order to participate. Infants who are started on antibiotic based on the presence of multiple non-red flags or one red flag, based on maternal or neonatal risk factors and do show elevated inflammatory parameters during the course of infection are also eligible for participation. See risk assessment tables in supplementary material.

Secondary outcomes

11. Throughout the manuscript there are discrepancies with regard to what is included in the secondary outcomes. Please address this and make uniform throughout. For example, on page 7 line 31 it states that duration of hospitalisation will be an included secondary measure, however this has not been stated elsewhere in the manuscript.

We do agree with the reviewer and have made the secondary outcomes identical throughout the manuscript.

12. Randomisation after 48 hours is the critical point in the RCT however, one suspects that in clinical practice there may be variation in how quickly participants are randomised to each group. This will not be an issue for the control group however it may be that certain participants in the intervention arm have more doses of IV antibiotics if the 48 hour switch to oral is not performed optimally in all participants. This should be considered by the authors and perhaps it would be useful to collect data with regard to the number of doses of both IV and oral antibiotics for each participant in the intervention arm, to record any deviations from the optimum 48 hour switch.

We do agree with the reviewer that the variance in timing of randomization should be addressed. Although we record timing of switch/administration of the first oral dose, registration of the number of IV doses before IV-to-oral switch is an interesting suggestion and we will therefore add this to the eCRF.

13. Given the RCT is not blinded to caregivers, parents and clinicians, the authors should address the standard procedure for if a parents wants their child to be in a specific group for the trial.

In the patient information form it is stated clearly that randomization is performed through a web-based program and cannot be influenced by the treated physician. If parents want their child to be in a specific group they will not be included for participation and their child will get standard treatment (iv). We have clarified this on page 8 of the manuscript.

14. The authors should consider the bias introduced by oral treatment being administered out of the hospital setting, to include some consideration of drug dose, duration and frequency compliance.

We are aware of the fact that since treatment is home-based, monitoring of medication administration differs from in hospital which might introduce a bias. If our study shows non-inferiority which might lead to a change in local guidelines, it will be daily practice for a child to receive home-based treatment. Assessing efficacy and safety in a controlled setting such as in hospital, where medication administration will be strictly monitored, we might over-estimate the efficacy and safety of the treatment. Moreover, we don't want to keep the infant hospitalized for treatment completion. Despite this, we are well aware of the potential bias and have tried to overcome this through the following study interventions:

1. Infants are only discharged when parents have shown to adequately dose and administer the antibiotic suspension.

2. Parents from infants in the intervention arm are phoned 1-2 days after discharge to evaluate medication administration, occurrence of side effects and overall well-being of the child.

3. Parents are asked to return the bottles with remaining antibiotic suspension at the final visit. We will then measure the remaining amount of suspension and evaluate if this corresponds to the total volume minus the treatment volume.

This is explained on page 10 of the manuscript.

15. Please could the authors include further details of the outcome survey to include the questions on signs of infection, side effects, quality of life and cost effectiveness. See page 9 line 12. Questionnaires have been added to the supplementary material.

16. Page 9, line 14 states the primary outcome in a different format to previously described (5 weeks, as opposed to 28 days), please standardise across the manuscript.

Corrected in manuscript on page 11.

17. Please could the authors ensure that all acronyms are adequately explained in full, for example GBS on page 9 and DSMB on page 9. And CEA on page 12.

Corrected in manuscript.

18. The DSMB are the allocated group who are to determine whether the primary outcome is a reinfection. It is important to state what factors will be considered in order to come to this decision.

Page 9

The study team defines whether an infection in the first 28 days after treatment completion is listed as a primary outcome. To facilitate decision making, a chart has been developed for the treating physician with a list of tests (including laboratory tests, cultures and viral diagnostics) that have to be performed when a child experiences a possible bacterial infection. All primary outcomes and all serious adverse events (thus, also covering all hospital admissions) are corresponded with the DSMB. They decide whether the adverse event is rightly classified as a primary outcome and perform an interim analysis if necessary.

19. The timing of the pharmacokinetic blood sampling, after the first 30 patients, is not defined. One assumes this is because the first 30 patients will establish the optimum time for blood sampling. However, it should be considered that the participant may be discharged home after the second dose of oral antibiotic, if tolerated. Therefore, if the optimum time for blood sampling is after the 3rd dose of oral antibiotic, this potentially prolonged hospital stay needs to be justified or alternative procedure should be put in place for the blood sampling.

If sampling occurs after the third administration this is because the 2nd administration is during nighttime and withdrawal is then more challenging since fewer laboratory assistants are available. Patients are never discharged during nighttime, therefore, in general, hospitalization will not be prolonged due to blood sampling. Apart from that, it is necessary for the newborn to stay a half to one more day after the antibiotic switch to evaluate if medication is correctly administered by parents. Extensive pharmacokinetic analysis (2 samples per patient) will be performed in the first 30 patients allocated to the intervention group and stopped once this number had been reached; therefore not leading to unnecessary prolongation of hospitalization.

20. With regard to the pharmacokinetic blood sampling, the authors should consider how this is a variable introduced into the intervention group but not the control group. Blood sampling is an invasive procedure and carries a small risk of introducing infection. The authors should consider how this may introduce bias and how this could be controlled for or at least acknowledged as a limitation.

Single blood sampling through a heel prick is common practice, among others for control of blood glucose- and bilirubin levels. To our knowledge this is not considered to be a risk factor for infection when performed properly. It holds true for the control group that when replacement or new insertion of

an intravenous cannula is needed, this can be a potential risk for infection. Fortunately, antibiotic treatment is broad and covers pathogens associated with catheter- or heel prick related infection such as Staphylococcus species. We collect data about the number of heel pricks and iv-cannula insertions. We have added this possible bias to the limitations. As mentioned in the comment above (19), pharmacokinetic sampling will be stopped after the first 30 patients.

21. The authors should state what demographic data will be collected to ensure the analysis and comparisons between groups is matched or controlled for.

As part of the quality of life and cost effectiveness questionnaires (added to the supplementary material) data regarding age of parents, education and background are collected. These data are not particularly important for our primary hypothesis but will be used for the secondary outcomes. We refer to the commentary on the study statistics for more information.

22. Storage of left over blood material. Please outline when the NGS will be conducted. In addition, please outline whether the NGS data will be available for DSMB when assessing whether a second infection is a reinfection or not.

This is a very interesting comment, and we have added some more info about NGS on the manuscript page 13.

23. Please provide as supplementary figure the gut microbiome questionnaire.

Questionnaires for microbiome sampling have been added to the supplementary material.

24. Please detail whether the microbiome data will be collected from the intervention arm only or both. The flow diagram seems to suggest both, however the text on page 11 does not.

Fecal samples will be collected in both groups, otherwise we could not answer the research question with respect to the microbiome development. This is clarified on page 13.

25. Please detail whether control microbiome samples will be collected for example from neonates with no antibiotic treatment.

A recent study conducted in the Netherlands, evaluated microbiome development in term newborn, including newborn exposed and not exposed to antibiotics in the first days of live (INCA study, (3). Once published, this could serve as a reference for the evaluation of our results.

26. With regard to cost effectiveness and productivity cost, please detail how the analysis will take into account prearranged maternity and paternity leave.

All working mothers are granted maternity leave in the Netherlands. Paternity leave is addressed in the questionnaires send on day 7 and 21. In our calculations of productivity losses, we will ignore prearranged, regular maternity and paternity leave. This is now briefly mentioned in the manuscript page 15.

27. Please provide further details of the quality of life questionnaire perhaps as a supplementary figure.

Quality of Life questions are integrated in the questionnaires send on day 7 and 21, which were added to the manuscript as supplementary material.

28. For the quality of life questionnaire the authors should consider whether questions with relation to parental anxiety may be important to include. For example, with regard to whether a certain type of intervention is more anxiety inducing and/or parental anxiety about future bacterial infection with relation to each intervention.

We agree on the fact that anxiety might be present in parents who have a newborn admitted to the hospital with an infection. Based on the clinical experience of the study team, we expect that anxious

parents, in general, will not consent to participation. Furthermore, we expect this to be more dependent of the parent, rather than of the type of intervention. Therefore, and to limit the respondent burden caused by the amount of surveys, we decided not to study parental anxiety. Of note, parents' quality of life and satisfaction will be analysed.

29. Ethics and dissemination – it may be important to consider whether there should be dissemination to the wider public and certainly public engagement with regard to this RCT. Given the importance of antibiotic stewardship, this could offer an opportunity for public engagement and discussion about antibiotic usage and consumption.

We do agree with the reviewer regarding the fact that public engagement could be beneficial for this study and for the discussion about antibiotic usage and consumption in general. As mentioned in the manuscript, The Dutch parent organisation “Vereniging van Ouders van Couveusekinderen (VOC)” was involved in the conceptualization of the trial and has given their approval for the execution of this study. They will publish the final results of this study on their website.

Next to that, one of the projectleaders, dr. Kornelisse, is responsible for the national guideline on neonatal infections and an active member of the Dutch Neonatal Infections Network, through which the results can be easily spread in order to promote implementation. We are currently working on public engagement by a website, research contests and regular newsletters. Moreover, if the final results of the study show non-inferiority of oral treatment, press releases will be sent after publishing the study outcomes in a peer-reviewed journal. We have expanded on this in the text on page 16-17.

Reviewer 2

A very worthwhile study that has the potential to collect extremely useful data relating not only to clinical management but also to the PK of oral antibiotics in neonates and the impact of oral antibiotics (especially a broad spectrum antibiotic such as amoxicillin/clavulanic) on the developing gastrointestinal microbiome. These are all questions that need answering to advance the practice of neonatal AMS. The role of WGS as complementary to blood culture in cases of neonatal culture negative sepsis is also important to define.

We thank dr. Cooper for reviewing our manuscript and are pleased to read that she thinks this study will potentially answer a number of relevant questions in the field of neonatal sepsis.

1. I would like to know why amoxicillin/clavulanic was chosen as the oral comparator. Do the authors have data on blood culture-proven neonatal sepsis that suggests that the spectrum is appropriate? Most common causative pathogens of early onset neonatal sepsis in the Netherlands are group B streptococci (GBS) and E. coli infections. Theoretically, also other streptococci, staphylococci and gram-negative Enterobacteriaceae could cause infections. The most common casual pathogen is definitely GBS, which is fully susceptible for amoxicillin. Since infections with Enterobacteriaceae or S. aureus cannot be fully excluded, we chose to add clavulanic acid. Majority of Enterobacteriaceae are still amox/clav susceptible in our region.

2. The dose of amoxicillin/clavulanic is higher than we use at our institution as we would give 22.5mg/kg bd rather than tds. This seems like a very high dose to me and might lead to a higher rate of adverse events.

It is true that we chose a relatively high dosage regimen for amoxicillin-clavulanic acid. The proposed regimen was adopted from the Dutch guidelines in which, for oral amoxicillin/clavulanic acid, a daily dose of 50/12.5 mg/kg/day divided in 3 doses is recommended. (4) For safety matters, and because of pharmacokinetic uncertainties, especially of clavulanic acid, we chose a higher dose, within the recommended range; a maximum amoxicillin dose of 90 mg/kg/d is also used in countries with reported resistance. We do agree that this might lead to side effects or adverse events which will be evaluated on a regular base at each follow up visit. The pharmacokinetic samples will give us more

information on the chosen dosage regimen and whether or not this leads to plasma levels within the therapeutic range.

With regard to the dosing frequency, it is advised to dose 3 times a day, especially in severe infections in order to reach a $T > MIC$ for the relevant pathogens. With BD dosing, especially taking at a lower dose, it is possible that $T > MIC$ of more than 50 % (advised for neonatal infections) is not reached.

Regarding the formal interim analysis - how will the DSMB determine whether the primary outcome is related to the index suspected bacterial infection or not? Since the index infection has not been proven (as a condition of inclusion of the patient in the trial) how can a determination be made regarding the relatedness of a second infection? Some more details here would be useful since this is the primary safety assessment for early cessation of the trial.

We agree with the reviewer that it is difficult to state whether or not a second infection should be considered as a re-infection, as previously mentioned by the first reviewer. We therefore refer to point 3 and 18 in the section above. Apart from that, we have mentioned this as a limitation on page 18.

4. Is there any other way of collecting blood from neonates for PK studies other than by collecting additional samples e.g. scavenging blood collected for routine clinical purposes?

We do indeed try to combine pharmacokinetic blood sampling with regular tests such as the national heel prick screening or bilirubin tests. Unfortunately, there is not always a need for extra regular tests if the baby is clinically well. In that case, additional sampling is necessary. We assume that this does not exceed the number of IV insertions needed in newborns in the control group; we collect data about these parameters as written above (number of iv cannula insertions and/or heel pricks).

5. Regarding microbiome alterations, why are samples only being collected from the patients receiving augmentin? Should samples be collected from the standard therapy patients as well? This was indeed not clearly mentioned in the manuscript as the other reviewer also mentioned, but fecal samples will be collected in both the intervention and control group. This has been corrected in the manuscript.

Reviewer 3

Authors presented a protocol for a balanced multicenter (17 large teaching hospitals in the Netherlands) randomized controlled trial assessing the non-inferiority of an early intravenous-to oral antibiotic switch compared to a full course of intravenous antibiotics in neonates (0-28 days of age) with a probable bacterial infection.

1. The sample size is calculated for randomized controlled trials not a multicenter randomized controlled trial. The multicenter nature of the study is ignored. Authors must estimate the design effect and multiply the calculated sample size (550) by the design effect.

We agree with the reviewer that, preferably, the multicenter nature of the study should be taken into account for the calculation of the sample size. However, we feel that it is impracticable for a number of reasons. The standard of care in participating hospitals is very similar; all follow the same national guideline on bacterial infections in newborns. We therefore do not expect that the effect of center will be large, however, lacking further information (which is not attainable) we feel that to guess at the specific design effect would be incautious, potentially resulting in an underpowered study. Further, the techniques available for the adaptation of sample size calculations to include a design effect are not readily applicable to our study. These techniques are nicely summarized in the recent (2018) paper by Markus Harden and Tim Friede, "Sample size calculation in multicenter clinical trials"⁽⁵⁾, and as detailed, do not cover scenarios wherein a non-inferiority hypothesis is evaluated for proportions. Incorporation of a design effect into our sample size would necessitate at the least complex simulations (for varied (guessed) design effect sizes), which is beyond the scope of this study. Finally, participation in this trial is not harmful to the patients and we are therefore in agreement that it is not

unethical to ignore the multicenter nature of the study (thereby potentially including more patients than we otherwise might). We propose instead the inclusion of a post-hoc report on the design effect.

The suggested statistical methods for analyzing the secondary outcomes (secondary outcomes will be analyzed using a t-test or Mann-Whitney U test and in the case of categorical data a Chi-square test will be used) are not appropriate because the multicenter nature of the study is ignored. Authors are familiar with the mixed effects models, however for both primary and secondary analyses mixed effect or Generalized Estimation Equation methods must be used.

We agree with the reviewer, and thank them for their comment. We feel this is best addressed separately for each of the secondary outcomes.

1. Pharmacokinetics: Analysis of the PK data will be largely descriptive. Population PK/PD models will be established with mixed effect models and center will be one of the variables or centers. (Page 13)
2. Microbiome: We are interested in both differences over time within patients and differences between patients at specific time points. We will therefore use mixed effect models for the analysis. These models will include adjustments for center and will include variables such as type of feeding, antibiotic use and the presence of siblings. We have expanded on the microbiome analysis in the text and added the questionnaire in the supplementary material. (Page 14)
3. Next Gen Sequencing (NGS): Initial analysis will be exploratory, and as such will be purely descriptive. If substantial differences are noted in the yield of recovered organisms from NGS compared to the blood culture, further analysis may be undertaken (exploring the relationship with clinical data). In that instance, mixed models will be used, and adjustments will be made for center, but also for inflammatory parameters at baseline and at 24 hours and clinical score. (Page 13)
4. Quality of Life: There is no validated survey to assess QOL in these patients and their parents. We have therefore created one, which includes variables which we feel provide pertinent information. The questionnaire is included in the supplementary material. Analysis of this data, which has an ordinary scale, will be exploratory and descriptive. If a trend is observed, mixed effect models can be used to further explore the effect of treatment and center. (Page 15-16)

References

1. Neuhaus TJ, Berger C, Buechner K, Parvex P, Bischoff G, Goetschel P, et al. Randomised trial of oral versus sequential intravenous/oral cephalosporins in children with pyelonephritis. *Eur J Pediatr.* 2008;167(9):1037-47.
2. Neonatal infection (early onset): antibiotics for prevention and treatment [Available from: Available from: <https://www.nice.org.uk/guidance/cg149>.
3. Rutten NB, Rijkers GT, Meijssen CB, Crijns CE, Oudshoorn JH, van der Ent CK, et al. Intestinal microbiota composition after antibiotic treatment in early life: the INCA study. *BMC Pediatr.* 2015;15:204.
4. Kinderformularium. Kinderformularium [Available from: <https://www.kinderformularium.nl/geneesmiddel/307/amoxicilline-clavulaanzuur>.
5. Harden M, Friede T. Sample size calculation in multi-centre clinical trials. *BMC Med Res Methodol.* 2018;18(1):156.

VERSION 2 – REVIEW

REVIEWER	Gabriella Morley University of Birmingham, United Kingdom
REVIEW RETURNED	11-Feb-2019

GENERAL COMMENTS	Many of the major points have been addressed and the RCT results will be a necessary addition to the body of scientific literature. Abstract - Thank you for clarifying the point on line 37 Introduction - Important RCT to conduct in order to address the gap in the research - More detail regarding evidence for early IV to oral switch has been included Methods and analysis - Thank you for providing details of participating hospitals - Primary outcome 9. Please further detail the inclusion criteria with specific emphasis on the raised CRP/PCT and clinical symptoms. For example what defines a raised CRP/PCT in this RCT? What clinical symptoms/signs define probably bacterial infection in this RCT? It may be appropriate to have a checklist of symptoms and to define how many of these symptoms or which symptoms in particular are necessary for the neonate to be included in the study. Discussion - Limitations sections has addressed the reinfection issue with regard to specific pathogen and the bias of the oral treatment administration Supplementary - Thank you for providing as supplementary figure the gut microbiome questionnaire Points not addressed - Randomisation after 48 hours is the critical point in the RCT however, one suspects that in clinical practice there may be variation in how quickly participants are randomised to each group. This will not be an issue for the control group however it may be that certain participants in the intervention arm have more doses of IV antibiotics if the 48 hour switch to oral is not performed optimally in all participants. This should be considered by the authors and perhaps it would be useful to collect data with regard to the number of doses of both IV and oral antibiotics for each participant in the intervention arm, to record any deviations from the optimum 48 hour switch. - The authors should state what demographic data will be collected to ensure the analysis and comparisons between groups is matched or controlled for.
---

REVIEWER	Celia Cooper Women's and Children's Hospital, North Adelaide, Australia
REVIEW RETURNED	30-Jan-2019

GENERAL COMMENTS	The authors have adequately addressed my initial comments either by revising the methods section or by acknowledging some inherent (not able to be corrected) weaknesses in the trial design. I am satisfied with the revised version of the research proposal. I wish the authors good luck with undertaking the study and I look forward to reading the results.
---

VERSION 2 – AUTHOR RESPONSE

Please further detail the inclusion criteria with specific emphasis on the raised CRP/PCT and clinical symptoms. For example what defines a raised CRP/PCT in this RCT? What clinical symptoms/signs define probably bacterial infection in this RCT? It may be appropriate to have a checklist of symptoms and to define how many of these symptoms or which symptoms in particular are necessary for the neonate to be included in the study.

As described in our previous response, the definition of “probable infection” is set by the treating physician, based on the inclusion criteria. These criteria are mentioned in table 1. Indication for prolonged antibiotic treatment despite negative cultures is, in our country, based on the national guideline which derives from the UK NICE guideline. A CRP ≥ 10 mg/L is considered to be elevated, for PCT we use the age-related normogram. (1) We have clarified this on page 7 of the manuscript. When prolonged antibiotic therapy is indicated and the patient meets the inclusion criteria, he/she can be included in the trial. We record all risk factors such as GBS status; duration of rupture of membranes; maternal fever and administered antibiotic therapy, and clinical symptoms, both at onset of suspicion and at T = 24, in our CRF. Those include respiratory, circulatory, gastro-intestinal, neurological symptoms, changes in temperature and need for support. The risk assessment tables which describe the clinical symptoms are available in the supplementary material.

1. *Randomisation after 48 hours is the critical point in the RCT however, one suspects that in clinical practice there may be variation in how quickly participants are randomised to each group. This will not be an issue for the control group however it may be that certain participants in the intervention arm have more doses of IV antibiotics if the 48 hour switch to oral is not performed optimally in all participants. This should be considered by the authors and perhaps it would be useful to collect data with regard to the number of doses of both IV and oral antibiotics for each participant in the intervention arm, to record any deviations from the optimum 48 hour switch.*

We agree with the reviewer that the variance in timing of randomization should be addressed and we have therefore adjusted our CRF. We record start of antibiotic therapy, date and time of switch, total number of IV administrations before switch, number of oral administrations and total duration of antibiotic therapy (iv + oral together). We have clarified this remark on page 8 of the manuscript.

The authors should state what demographic data will be collected to ensure the analysis and comparisons between groups is matched or controlled for.

As part of the quality of life and cost effectiveness questionnaires (added to the supplementary material) data regarding age of parents, education and background are collected. We have clarified which data we collect on page 9 of the manuscript.

VERSION 3 – REVIEW

REVIEWER	Gabriella Morley University of Birmingham, United Kingdom
REVIEW RETURNED	27-Apr-2019
GENERAL COMMENTS	The authors have updated the manuscript to improve upon it in the manner that I previously suggested.